# The Efficiency of Selected Green Solvents and Parameters for Polyphenol Extraction from Chokeberry (*Aronia melanocarpa* (Michx)) Pomace

**DOI:** 10.3390/foods12193639

**Published:** 2023-10-01

**Authors:** Efaishe Tweuhanga Angaleni Kavela, Lilla Szalóki-Dorkó, Mónika Máté

**Affiliations:** Department of Fruits and Vegetables Processing Technology, Institute of Food Science and Technology, Hungarian University of Agriculture and Life Sciences, Villányi Street 29-43, H-1118 Budapest, Hungary; kavela.efaishe.tweuhanga.angaleni@phd.uni-mate.hu (E.T.A.K.); mate.monika.zsuzsanna@uni-mate.hu (M.M.)

**Keywords:** chokeberry, pomace, solvents, extraction conditions, polyphenols, antioxidants

## Abstract

Chokeberry pomace is a potential source of natural colourants, antimicrobial agents, and anti-senescence compounds due to its high polyphenols content. Therefore, this study assessed the efficiency of green solvents (50% ethanol, 50% glycerol, and 100% distilled water, all acidified with 1% citric acid or 1% formic acid) for anthocyanin and total phenolic content (TPC) extraction from lyophilised chokeberry pomace. Extraction was performed in a water bath at 40, 50, and 60 °C for 60 and 120 min, followed by ultrasonic treatment for 15 and 30 min. Based on the results, 50% ethanol + 1% citric acid yielded significantly higher total anthocyanin content in the case of both spectrometric and HPLC measurements (1783 ± 153 mg CGE/100 g DW and 879.5 mg/100 g DW) at 50 °C for 60 min. Citric acids seem more effective compared to formic acid. The highest values of TPC were obtained with 50% glycerol + 1% formic acid at 50 °C for 60 min (12,309 ± 759 mg GAE/100 g DW). This study provides evidence that a substantial quantity of polyphenols, which can potentially be used as a natural food additive, can be efficiently extracted with 50% ethanol + 1% citric acid or 50% glycerol at 50 °C for 60 min from chokeberry pomace.

## 1. Introduction

The use of fruit by-products has been growing rapidly since they have been identified as abundant in polyphenols [1,2]. The use of by-products is also encouraged in many countries, including Hungary, in support of sustainable food processing [3]. This includes the use of chokeberry pomace, a residual left after juice extraction from chokeberries [3,4]. Chokeberry (*Aronia melanocarpa* (Michx)) is a shrub belonging to the Rosaceae family, originally from East Canada and the eastern part of North America [5]. It started spreading in European countries 100 years ago, where fruits are processed into juice, wine, jam, and tea [6]. This fruit has been reported to be abundant in polyphenols, specifically phenolic compounds, and anthocyanins [5,7]. However, a large amount of these compounds is retained in the pomace compared to the juice [4,8,9]. Traditionally, fruit pomace is used in animal feed, organic fertilizers, or biofuel production or rather discarded [10]. In such cases, the polyphenols contained in the pomace go to waste. Polyphenols are vital in food for their antioxidant properties which eliminate free radicals, consequently maintaining the natural appearance and the microbial quality of food for the expected period [11,12]. Food colour is the most important aspect which influences the consumer’s perspective about a food product, hence the addition of colourants to many foods in the market. Anthocyanins, one of the compounds richly found in chokeberries, are responsible for the red-dark pigment in the chokeberry pomace. This compound can be extracted from the pomace and be used in food as a cheap natural colourant. Therefore, chokeberry extracts have the potential to be used in the food industry as natural food additives. 

The recovery of polyphenols from the fruit pomace or any other material begins with extraction, which can be performed using different methods, solvents, and extraction parameters. Different extraction methods including solvent extraction, ultrasound-assisted extraction, enzyme-assisted extraction, and supercritical extraction have been used to isolate polyphenols from chokeberry fruits and pomace [13,14,15,16]. Solvents, namely ethanol, water, methanol, glycerol, and acetone, have been used in several studies [11,14,17]. Generally, the solvent to be used is selected based on the chemical nature of the polyphenols to be analysed [18]. Polyphenols range from simple molecules to highly polymerised compounds and can be found in complexes with other food components [19]. Hence, they differ from each other in polarities and solubility [20]. Therefore, the optimal solvent would be the one in which the targeted polyphenols can dissolve. In addition, extraction efficiency may also be influenced by other factors such as the sample particle size, pH, extraction duration (time), and temperature [20,21].

Most solvents used for solvent extraction are acidified with low concentrations of strong acids like hydrochloric acid [9] or weak acids such as formic acid, acetic acid, or citric acid [22,23,24]. The acidic condition helps polyphenols to be more stable and neutral. It also facilitates hydrolysis, consequently releasing polyphenols that are bound to other compounds [19]. According to Le et al. [25], the addition of citric acid to aqueous solutions resulted in better colour stability, less degradation, and lower anthocyanin content compared to solutions that were not acidified. In a study by Ćujić et al. [14], total anthocyanin content was extracted from dried chokeberries using a maceration technique with varying ethanol-water concentrations (100% distilled water, 50%, 70%, and 96% *v*/*v*). The study yielded high anthocyanin content with 50% *v*/*v* ethanol-water concentration. In addition, this same study also proved differences in the anthocyanin content yielded from samples extracted with different material/solvent ratios (1:10, 1:20, and 1:30), extraction temperatures (30–80 °C), and duration (15–90 min).

The factors stated above are an indication that there is a need for further study to determine the optimal extraction solvent and other parameters to yield a promising quantity of valuable compounds from chokeberry pomace. Looking at different studies that have extracted polyphenols from chokeberry pomace [14,15,16], different solvents and conditions have been used, giving information on the possible extraction methods, solvents, temperature, and time. However, little information is available on the optimal solvents and extraction conditions to obtain a significant quantity of polyphenols, more exactly anthocyanins. Therefore, this study focused on assessing the performance of different solvents under different extraction conditions to select the extraction solvent and parameter combination yielding a promising quantity of anthocyanins from chokeberry pomace for further utilisation as natural colourants.

## 2. Materials and Methods

### 2.1. Plant Material

The chokeberry fruits (Nero variety) were collected from a farm near Lajosmizse (47°02′44.4″ N 19°35′14.8″ E), Hungary in 2022 during the full ripening stage. The sample was packaged and stored at −18 °C until further processing.

### 2.2. Reagents and Standards

For extraction, ethanol (96% P.A), glycerol (87% P.A), formic acid, citric acid, and distilled water were used. For the measurements, methanol, Folin-Ciocalteu phenol reagent (Sigma Aldrich, St. Louis, MO, USA), sodium acetate (pH 4.5 buffer), potassium chloride (pH 1.0 buffer), hydrochloric acid, and anhydrous sodium carbonate (Na_2_CO_3_) (Riel-del Haen, Germany) were used.

The HPLC Gradient-grade acetonitrile and water were purchased from Fisher Scientific (Loughborough, UK). Formic acid was obtained from Fluka (Buch, Switzerland). Cyanidin-3-glucoside, cyanidin-3-arabinoside, and cyanidin-3-galactoside, as well as gallic acid standards, were purchased from Merk (Darmstadt, Germany).

### 2.3. Samples Preparation

The chokeberry fruits were processed into juice and the pomace using pressing technology under laboratory conditions. After this step, the pomace was lyophilised using a Leybold Heraeus Lyovac CT2 freeze dryer (Labexchanger, Burladingen, Germany). The moisture content of dried pomace was 11.77%, while the water activity decreased to 0.221. The lyophilized pomace was then grounded into 2.2 mm powder with a hand blender (Minichiller 300 (Huber, Germany)). Approximately 300 mg of the lyophilised pomace was weighed into centrifuge tubes and 9 mL solvents and mixed thoroughly. The solvents used were ethanol-distilled water (50:50% *v*/*v*), glycerol-distilled water (50:50% *v*/*v*), and distilled water (100% *v*/*v*), all acidified with 1% citric acid or formic acid in parallel (Table 1).

### 2.4. Extraction Procedures

The prepared samples were placed in the water bath with a controlled temperature, where the extraction was carried out at different temperatures (40, 50, and 60 °C) and durations (60 and 120 min) as indicated in Table 1. For further extraction, samples were transferred to the ultrasonic (Bandelin, RK 52) and sonicated for 15 or 30 min at 35 kHz (as indicated in Table 1) to determine the optimal duration of the ultrasonic-assisted extraction. After sonication, samples were centrifuged at 4500 rpm for 5 min in a Z206A laboratory centrifuge (Hermle, Germany). After centrifugation, the supernatant was collected and used for total anthocyanins (TA), total phenolic content (TPC), colour parameters determination, and high-performance liquid chromatography (HPLC) analysis. The extraction and all tests were carried out in triplicate. Before the HPLC measurement, the supernatant was filtered through a 0.45 µm membrane filter before injecting 10 µL into the HPLC system.

### 2.5. Total Phenolic Content (TPC)

The amount of total phenolic content was determined by following the colourimetric method, as described by Singleton and Rossi [26]. The absorbance of the samples was measured against the blank solution at 760 nm with a spectrophotometer (Hitachi U-2900, Budapest, Hungary). The results are given in mg gallic acid equivalents per 100 g (mg GAE/100 g).

### 2.6. Total Anthocyanin Content

The total anthocyanins contained in the chokeberry pomace extract samples were determined by using a pH differential method as described in the Association of Official Agricultural Chemists (AOAC) Method 2005:07 [27]. The absorbance was measured spectrophotometrically against the blank sample at 520 nm as well as at 700 using pH 1 and pH 4.5 buffer solutions. The obtained results were reported as mg cyanidin 3-glucoside equivalents per 100 g of pomace extract sample weight (mg CGE/100 g).

### 2.7. High-Performance Liquid Chromatography (HPLC) Analysis

Based on the pH differential method, ten (10) extracted samples that gave the best anthocyanin yield were analysed further with HPLC for qualitative and quantitative results. Chromatographic separation of anthocyanin components was performed with a Shimadzu HPLC system (Shimadzu Corporation, Kyoto, Japan) using a 150 × 4.6 mm, C18, 3 µm particle size column (Phenomenex Torrance, California, USA). For the elution, 0.5% (*v*/*v*) formic acid in high-purity water (mobile phase A) and 0.5% formic acid in acetonitrile (mobile phase B) were used as solvents at a low flow rate of 0.5 mL min^−1^. The total gradient program was 30 min and started at 55 solvent B. Solvent B was increased linearly to 25% in 5 min, and from 5 to 10 min, solvent was increased to 100. Finally, 100% solvent B was held constant for another 5 min, then it was immediately reduced to 5% and the system was run for an additional 10 min at the initial solvent composition. Samples were analysed at 520 nm. Qualification of different peaks was carried out using the retention time of anthocyanin standards, while the quantification was by four-point external calibration with cyanidin-3-arabinoside (CA); the results were expressed in mg cyanidi-3-galactoside equivalent CGaE/100 g DW.

### 2.8. Colour Measurement

Since chokeberry pomace is a rich source of anthocyanins which are responsible for the dark red colour in the chokeberry pomace, the colour intensity between different samples was measured with a digital colourimeter (Konica Minolta, Chroma-400). This collurimeter determines the lightness (L*, 0-100), redness (+a*)/greenness (“-“a*), and yellowness (+b*)/blueness (“-“b*) of the sample. In this case, the colour change was only calculated between formic acid and citric acid samples of 50% ethanol extracts because ethanol had given a high anthocyanin yield. The 50% ethanol (*v*/*v*) and 50% glycerol (*v*/*v*) samples were selected. To determine the colour change between samples, the overall change (∆E*) was calculated as follows:E* = √(〖∆L〗^(*2) + 〖∆a〗^(*2) + 〖∆b〗^(*2)),(1)
the noticeability in the colour change (∆E*) was defined as classified by Lukács [28] as follows: 0–0.5 = not noticeable, 0.5–1.5 = slightly noticeable, 1.5–3.0 = noticeable, 3.0–6.0 = clearly visible, and 6.0–12.0 = great visibility.

### 2.9. Statistical Analysis

The results obtained were statistically analysed with IBM SPSS statistics software, version 27 (IBM Corp., New York, NY 10022, USA, 2020). The mean differences between factors were analysed using the one-way analysis of variance (ANOVA) post-hoc test (Turkey’s, Games’ Howell). Kolmogorov-Smirnova (*p* > 0.05), as well as Skewness and Kurtosis (Absolute value of Skewness and Kurtosis = 2 and 4 respectively), were used to prove the normality of error for different analyses. To ensure accurate analysis, the homogeneity of variances between the mean values of parameters was tested using Levene’s test. If the *p* value was greater than 0.05, the homogeneity was confirmed. However, if the *p* value was less than 0.05, the maximum variance/minimum variance test was conducted to determine the severity of the violation of homogeneity between parameters. The significant difference between factors was determined at the interval level of *p* < 0.05.

## 3. Results

To evaluate the efficiency of extraction parameters and solvents (50% ethanol, 50% glycerol, and 100% distilled water, all acidified with 1% citric or 1% formic acid) in extracting total anthocyanins (TA), total polyphenols content (TPC), HPLC, and colour parameters (L*, a*, and b*), the results obtained via spectrophotometry were calculated and then statistically analysed. The mean values were then compared between the different samples to determine the potential optimal solvents, temperature, and time for valuable compound extraction from the chokeberry pomace.

### 3.1. The Efficiency of Extraction Solvents on TPC and TA

The efficiency of extraction solvents (50% ethanol-formic acid, 50% ethanol-citric acid, 50% glycerol-formic acid, 50% glycerol citric acid, 100% distilled water + 1% formic acid, and 100% distilled water-citric acid) at different extraction temperatures and durations on TA and TPC is shown in Figure 1A,B, respectively. In addition, the total yield of both TA and TPC between solvents, temperature, and extraction duration were compared as indicated in Table 2. The maximum levels of TA were obtained mostly from samples extracted with 50% ethanol + 1% citric acid (ranging from 1368 ± 234 to 1783 ± 154 mg CGE/100 g DW). The 50% ethanol + 1% citric acid indicated a total yield of TA significantly higher (*p* < 0.05) compared to all other five (5) solvents, including 50% ethanol + 1% formic acid (Table 2). This was followed by 50% ethanol + 1% formic acid yields (ranging from 1362 ± 145 to 1767 ± 225 mg CGE/100 g DW) and 50% glycerol + 1% formic acid yields (ranging from 1469 ± 109 to 1573 ± 67 mg CGE/100 g DW). The 100% water extracts gave significantly the lowest (*p* < 0.05) yield. The highest yield, which was 1783 ± 154 mg CGE/100 g DW, was obtained with 50% ethanol + 1% citric acid at 50 °C for 60 min. This yield was significantly (*p* < 0.05) higher than the TA levels obtained from 50% ethanol-citric acid extracts at 40 °C for 120 min and 60 °C for both 60 and 120 min (Figure 1A).

Based on the results (Figure 1A,B), the solvents’ effects varied between TPC and TA. The maximum TPC yields were achieved using 50% glycerol + 1% formic acid or 50% ethanol + 1% citric acid. The yields ranged from 9188 ± 583 to 12,308.5 ± 758 mg GAE/100 g DW and from 8540 ± 788 to 10,201 ± 836 GAE/100 g DW, respectively (as shown in Figure 1B), whereas the highest TA yields were given by 50% ethanol + 1% citric acid. Based on the overall yield of TPC per solvent, 50% glycerol + 1% formic acid gave significantly higher (*p* < 0.05) results compared to other solvents (Table 2). The highest yield was obtained at 50 °C for 60 min, but it was not significantly different (*p* > 0.05) from the results of 50% glycerol-formic acid obtained at 50 °C for 120 min and 60 °C for 60 min (as shown in Figure 1B). The extraction with 100% water solvents yielded the lowest TPC and TA values.

### 3.2. The Efficiency of Extraction Temperature and Time

The effects of extraction temperature and duration on TA and TPC in this study are shown in Figure 2A,B, respectively. The significant differences in the overall yield of TA and TPC between extraction temperatures and durations are indicated in Table 2. Based on the TA results (Figure 1A and Table 2), the good results given by 50% ethanol with 1% citric acid were obtained at 50 °C, followed by 60 °C and then 40 °C, for both 60 and 120 min. These results did not differ significantly (*p* > 0.05) between the extraction temperatures and durations. Similar results indicating no significant differences (*p* > 0.05) between the effects of the extraction temperatures and durations were obtained in TPC yield (Figure 1 and Table 2). However, the highest yields of TPC were obtained at 60 °C and 50 °C, followed by 40 °C, for both 60 and 120 min, with no significant difference (*p* > 0.05) indicated between these results. Based on the results, the extraction duration effects (60 and 120 min) on both TA and TPC do not significantly differ (*p* > 0.05). 

### 3.3. Colour Parameters of Ethanol Extracts

The good yield of anthocyanins was obtained with 50% ethanol + 1% citric acid, followed by 50% ethanol + 1% formic acid. Therefore, the colour differences (L*, a*, and b*) were calculated between 50% ethanol + 1% citric acid and 50% ethanol + 1% formic acid samples extracted at 40 and 50 °C for both 60 and 120 min, which had given high anthocyanins yield. The results are indicated in Table 3. The colour differences between 50% ethanol + 1% citric acid and 50% ethanol + 1% formic acid samples were interpreted based on their noticeability, according to Lukács’ [28] interpretation. Based on the results (Table 3), there was no noticeable difference in colour between 50% ethanol + 1% citric acid and 50% ethanol + 1% formic acid samples. However, the change in colour was slightly noticeable on samples extracted at 40 °C for 60 min in the water bath and 15 min in the ultrasound; however, it was not a significant change.

### 3.4. Identification and Quantification of Anthocyanins in the Chokeberry Pomace

Based on the TA spectrophotometer method, ten (10) samples that had the highest total anthocyanin content were selected and further analysed with HPLC to identify and quantify the main three cyanidin glycosides, namely cyanidin 3-galactoside, cyanidin 3-glucoside, and cyanidin 3-arabinoside [29,30]. The result of the HPLC chromatogram of the chokeberry extracts recorded at 520 nm is shown in Figure 3. Whereas the quantitative and qualitative HPLC results of anthocyanins are given in Table 4. The main targeted anthocyanins were identified based on their retention time and available standards.

Based on the results (Figure 3), the highest peak (peak 1) was identified as cyanidin 3-galactoside, which eluted first before all other peaks and was then followed by cyanidin 3-glucoside (peak 2), which indicated the lowest peak in comparison to the first two peaks. Cyanidin 3-arabinoside was identified as peak 3. There is a fourth peak, which presumably is cyanidin 3-xyloside [29]. The all chromatogram can be seen in the Appendix A (Appendix A).

The average sum of individual anthocyanins given by the selected samples ranges from 716.41 to 938.66 mg CGaE/100 g DW. The highest total anthocyanin yield (938.6 mg CGaE/100 g DW) measured with HPLC analysis was obtained from extracts of 50% ethanol + 1% citric acid at 50 °C for 60 min in the water bath and 15 min in the ultrasound. This was followed by 924.57 mg CGaE/100 g DW from the extracts of 50% ethanol + 1% citric acid at 40 °C for 60 min in the water bath and 15 min in the ultrasound. The third highest yield (918.28 mg CGaE/100 g DW) was also obtained with 50% ethanol at 50 °C for 60 min and 30 min in the ultrasound. These results correspond with the total anthocyanin yields obtained with the pH differential method, where samples extracted with 50% ethanol + 1% citric acid were reported to have the highest yield compared to other solvents. In addition, samples extracted with 50% ethanol + 1% citric acid at 50 °C in the water bath for 60 min were also reported in the pH differential method with the highest yield. HPLC results also indicate that citric acid results give higher yield than that of formic acid extracts. As discussed earlier, 50% ethanol and 50 °C have been reported in the literature to be optimal for polyphenol extraction from chokeberry pomace.

As regards the anthocyanin profile of extracts, the most abundant pigment compound is cyanidin-3-galactoside. Its concentration ranged between 485.57 ± 61 and 597.05 ± 41 mg CGaE/100 g DW). Cyanidin-3-glucoside was only detectable by HPLC in case of extraction with citric acid; it was present in similar amount as the presumably identified cyanidin-3-xyloside. The second major anthocyanin was cyanidin-3-arabinoside with 197.13 ± 23 to 262.69 ± 34 mg CGaE/100 g DW. Based on the result, it can be concluded that the different extraction methods had no selective effect on the individual pigments; however, using citric acid is recommended to obtain higher anthocyanin concentrations during the extraction.

## 4. Discussion

The efficiency of the selected green solvents and parameters for the extraction of TPC and TA from chokeberry pomace were assessed. The 50% ethanol + 1% citric acid dominated with high yields, followed by 50% glycerol + 1% formic acid. The 100% water gave the lowest in all tests. In a study by Sik et al. [31], 50% ethanol was reported as an optimal solvent for polyphenol extraction from fruits compared to different ethanol-water mixture ratios. The efficiency of 100% water extraction on polyphenols was less, as we expected. The low yield of polyphenols from water extracts has been reported in a study by Thi and Hwang [32], where polyphenols were extracted from Aronia leaves using 100% distilled water and 80% ethanol.

In this study, it was not only 50% ethanol that gave good yield, but 50% glycerol also demonstrated efficient extraction, especially in TPC. This is an indication that 50% glycerol can be an alternative extraction solvent. These results align with the results obtained in a study by Kowalska et al. [17], in which the usage of 50% glycerol as an extraction solvent for anthocyanins from black chokeberry and elderberry fruits was found efficient at 20 °C and 50 °C. Ethanol, glycerol, water, and citric acid are all biodegradable, recyclable, non-volatile, time- and energy-efficient, and environment-friendly [33]. Extracting for a shorter period can prevent the degradation of polyphenols that may occur during a longer extraction duration. In a study by El Kantar et al. [34], the use of green solvents has been proven to improve energy efficiency. Furthermore, these solvents are not like hydrochloric acid, formic acid, or traditional solvents that are volatile, flammable, and toxic; they do not have any harmful effects on human health or safety [35].

To obtain the maximum yield of polyphenols, the polarity of compounds and solvents to be used should be considered. Water might be the safest solvent; however, it cannot be efficiently used as an extraction solvent alone. This is because polyphenols can be bound to different biomolecules including inorganic compounds, polysaccharides, and proteins found in the material, which have different solubility and may not all be soluble in water or alcohol alone [19,30]. Therefore, water needs to be mixed with other solvents like ethanol or glycerol plus an acid to facilitate the hydrolysis and solubility of compounds for the maximum yield. The polyphenol compounds found in different fruits have different biochemical natures and polarities; hence, their solubility in different solvents is different as well [36]. Thus, the selection of solvents to be used should be done according to the type of polyphenol in the target. Chokeberries have phenolic compounds, anthocyanins, and various flavonol classes that can dissolve in either water or alcohol [29]. Ethanol, glycerol, and water are polar green solvents, which makes them suitable for extraction of TPC, TA, and antioxidant capacity from chokeberry pomace. However, they should be in the appropriate ratio with water to achieve a significant yield.

Suitable extraction should also be used at the optimal temperature. The efficiency of extraction temperatures differs between the components being extracted, and it also depends on the duration of exposure to temperatures. Reflecting on the overall results, regardless of other factors influencing the extraction yield, temperatures of 50 °C for 60 min have given good yield on all three (3) tested parameters. The extraction temperature is a crucial factor to consider when extracting polyphenols. By raising the temperature, the pomace tissues can be softened and the linkages between polyphenols and their associated components are weakened [37], consequently reducing the pomace’s surface tension and enhancing the polyphenols’ solubility and mass transfer [37]. This would hence result in maximized yield. However, the results may differ between solvents.

Although elevated temperatures might function as an aid in maximizing yield, high temperatures can have negative effects on the yield of polyphenols. In a study by Palma et al. [37], temperatures from 50 to 100 °C were reported to have increased the phenolics yield, and temperatures above 100 °C were reported with a declined yield. These results are in line with a study by Le et al. [25] which also reported a decline in anthocyanin concentration (from *Carissa carandas* L. fruits) as the temperature increases above 50 °C. The same study also reported a drop in anthocyanin yield after reaching the “peak” while still exposed to high temperatures. This may not be the case in this study because there are no significant differences between 60 °C and 120 °C results. However, it is worth noting that prolonged exposure of polyphenols to raised temperatures can decrease yield.

The selected extraction temperatures, durations, and ethanol acidified with 1% ethanol, or 1% formic acid have demonstrated positivity in extracting colour/anthocyanins from dry chokeberry pomace. Anthocyanins are responsible for the pigmentation in the chokeberry. As a natural colourant, their use has attracted much interest in the food industry for the aqueous food system [38]. However, its stability is a concern because it is affected by many factors including pH, and hence, it can easily be degraded, especially at pH values above 7 [30]. In a study by Amelia et al. [39], 70% ethanol acidified with 3% citric acid was found more effective in extracting high yields of anthocyanins and colour intensity. However, due to anthocyanins’ instability, pH values from 4–5 were identified as better for anthocyanins than higher pH values. This could mean that, although 1% citric or formic acid and 50% ethanol have been found to be a good combination in this study for anthocyanin extractions, further study might be needed to prove the anthocyanins’ stability. Lin et al. [40] in their review suggested modification of technologies to improve the anthocyanins’ stability. In a study by Abou-Arab et al. [41], both ethanol acidified with 1% citric acid and 2% citric acid were used, and it was discovered that 2% ethanol gives higher colour intensity than 1% citric acid.

However, based on this study and its aim, the 50% ethanol acidified with 1% citric acid indicated efficiency compared to the other used solvents, especially at a water bath extraction temperature of 50 °C. This is indicated by both HPLC and differential method results. Moreover, the anthocyanin composition obtained with HPLC is in line with the results obtained from chokeberry pomace by Oszmiański and Wojdylo [9]. However, their results are slightly higher than those obtained in this study because they depend on several factors, including the extraction method as well as the cultivar used. This study’s HPLC patterns are also in accordance with patterns reported by several studies, in which cyanidin-3-galactoside eluted first and had a higher peak than the rest of the cyanidins [9,29,41]. The identification of the peaks aligns with the quantification results indicated in Table 4. All samples indicated a high yield of cyanidin-3-galactoside ranging from 485.6 ± 61.0 to 597.1 ± 41.1 CAE mg/100 g DW.

## 5. Conclusions

The high concentration of polyphenols found in chokeberry pomace has sparked interest in using it as a natural food additive. However, it is crucial to ensure that the extraction process is efficient in terms of yield, polyphenol stability (such as that of anthocyanins), and cost. This requires careful selection of extraction solvents and other parameters, particularly since polyphenols belong to different classes with varying natural structures. This study investigated the impact of solvents, extraction temperature, and solvent pH level on polyphenol yield to determine their effectiveness. For the extraction of TPC and TA, three solvents, namely 50% ethanol, 50% glycerol, and 100% water acidified with 1% formic acid or 1% citric acid, were utilized. The extraction process involved exposing the samples to different water bath extraction temperatures (40, 50, and 60 °C) for 60 or 120 min, with the addition of ultrasonic assistance for 15 or 30 min. Based on the overall results, 50% ethanol acidified with 1% citric acid and water bath extraction at the temperature of 50 °C for 60 min have been found optimal for extraction of anthocyanins. Whereas 50% glycerol acidified with 1% formic acid has been identified as optimal for extracting TPC at both 50 and 60 °C for either 60 or 120 min. This study also concludes that 50% glycerol acidified with 1% citric acid can be used for extraction of polyphenols at 60 °C in the water bath for 60 min, in replacement of ethanol. Although 50% ethanol gave the highest anthocyanin results, there were no noticeable colour changes calculated between 50% ethanol acidified with 1% formic acid and 5% ethanol with 1% citric acid. The green solvents used in this study indicate time and extraction efficiency, and they do not pose any danger to the environment or human health and safety risks. Therefore, they can be used to replace conventional solvents which are not environmentally friendly, pose human health and safety risks, are time-consuming, and have low extraction efficiency.

## Figures and Tables

**Figure 1 foods-12-03639-f001:**
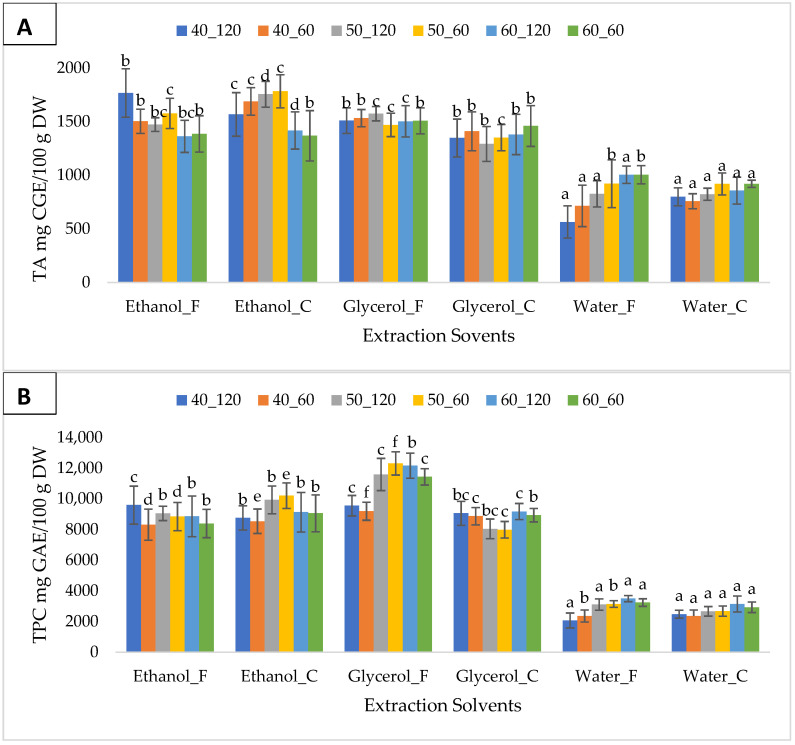
Extraction performance of six different solvents at different temperatures and durations on (**A**) total anthocyanins and (**B**) total phenolic content from freeze-dried chokeberry pomace. Letters indicate significant differences (0.95 confidence interval) between extraction solvents for each parameter. Error bars = standard deviation. C = citric acid, and F = formic acid.

**Figure 2 foods-12-03639-f002:**
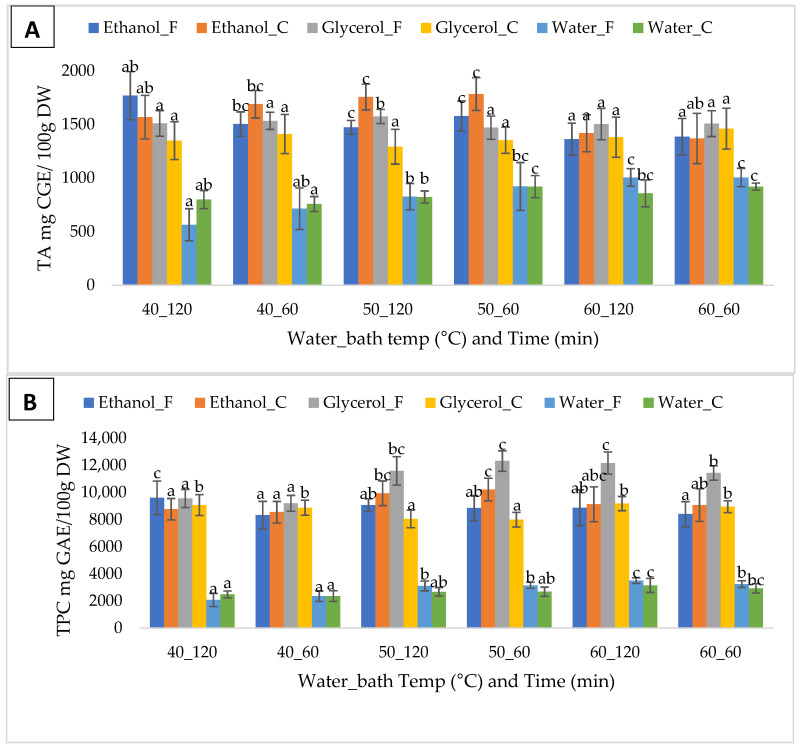
Extraction performance of six different solvents at different temperatures and times on (**A**) total anthocyanins and (**B**) total phenolic content from freeze-dried chokeberry pomace. Letters indicate significant differences (0.95 confidence interval) in extraction parameters for each solvent. Error bars = standard deviation.

**Figure 3 foods-12-03639-f003:**
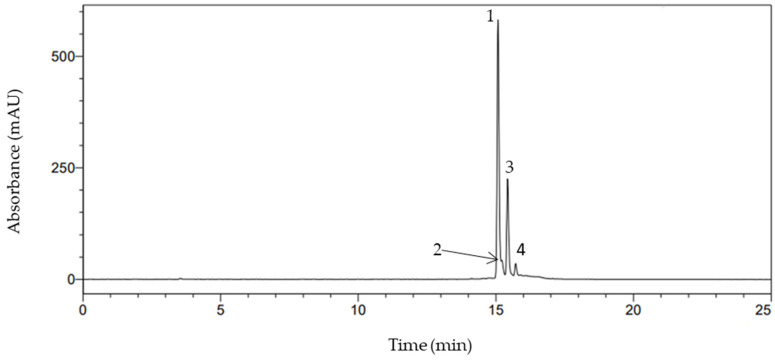
HPLC profile of chokeberry pomace detected at 520 nm. Peak 1 = cyanidin-3-galactoside, 2 = cyanidin-3-glucoside, 3 = cyanidin-3-arabinoside, and 4 = cyanidin-3-xyloside.

**Table 1 foods-12-03639-t001:** Parameters and solvents used in polyphenol extraction from lyophilised chokeberry pomace.

Water Bath		Ultrasound Duration	Solvents with Acidifiers
Temperature (°C)	Duration (min)	Duration (min)	
40	60	15 30	Ethanol-Distilled water (50:50% *v*/*v*) (1% citric acid) Ethanol-Distilled water (50:50% *v*/*v*) (1% formic acid) Glycerol-Distilled water (50:50% *v*/*v*) (1% citric acid) Glycerol-Distilled water (50:50% *v*/*v*) (1% formic acid) 100% Distilled water (1% citric acid) 100% Distilled water (1% formic acid)
	120	15 30	
50	60	15 30	
	120	15 30	
60	60	15 30	
	120	15 30	

**Table 2 foods-12-03639-t002:** The significant differences between the effects of solvents, temperature, and duration on TA and TPC yield.

	The Significant Differences between the Effects of Extraction Solvents					
Solvents	Ethanol + Formic acid	Ethanol + Citric acid	Glycerol + Formic acid	Glycerol + citric acid	Water + Formic acid	Water + citric acid
TA mg CGE/100 g DW	1511 ± 201 ^c^	1596 ± 234 ^d^	1515 ± 112 ^c^	1373 ± 176 ^b^	839 ± 218 ^a^	846 ± 101 ^a^
TPC mg GAE/100 g DW	8844 ± 1078 ^b^	9268 ± 1136 ^c^	11036 ± 1432 ^d^	8676 ± 746 ^b^	2905 ± 607 ^a^	2712 ± 448 ^a^
	The Significant Differences between the Effects of Extraction Temperature					
	40	50	60			
TA mg CGE/100 g DW	1263 ± 436 ^a^	1313 ± 363 ^a^	1264 ± 275 ^a^			
TPC mg GAE/100 g DW	6762 ± 3245 ^a^	7463 ± 3514 ^a^	7496 ± 3309 ^a^			
	The Significant Differences between the Effects of Extraction Duration					
	60 min	120 min				
TA mg CGE/100 g DW	1293 ± 350 ^a^	1268 ± 378 ^a^				
TPC mg GAE/100 g	7155 ± 3331 ^a^	7325 ± 3411 ^a^				

Letters indicate significant differences (0.95 confidence interval) between extraction solvents for each parameter. Error bars = standard deviation.

**Table 3 foods-12-03639-t003:** The colour differences between samples with 50% ethanol (formic and citric acid) extracted at 40 and 60 °C.

Water Bath (°C)	Time (min)	Ultrasonic (min)	50% Ethanol	Mean			Colour Change	
				L*	a*	b*	∆E*(Citric − Formic Acid)	Evaluation
40	60	15	1% formic acid	19.07	1.8	1.67	0.59	slightly noticeable
			1% citric acid	19.32	1.29	1.52		
40	60	30	1% formic acid	19.21	1.36	1.48	0.13	not noticeable
			1% citric acid	19.34	1.39	1.46		
40	120	15	1% formic acid	19.46	1.64	1.54	0.28	not noticeable
			1% citric acid	19.41	1.37	1.51		
40	120	30	1% formic acid	19.38	1.21	1.38	0.15	not noticeable
			1% citric acid	19.51	1.22	1.45		
50	60	15	1% formic acid	19.41	1.4	1.41	0.18	not noticeable
			1% citric acid	19.38	1.23	1.44		
50	60	30	1% formic acid	19.29	1.47	1.61	0.38	not noticeable
			1% citric acid	19.27	1.12	1.47		
50	120	15	1% formic acid	19.11	1.41	1.52	0.23	not noticeable
			1% citric acid	19.23	1.22	1.46		
50	120	30	1% formic acid	19.34	1.32	1.54	0.38	not noticeable
			1% citric acid	19.33	0.96	1.41		

**Table 4 foods-12-03639-t004:** Individual anthocyanin concentration after extractions.

Anthocyanin Concentration (mg CGaE/100 g DW)								
Solvent	Water-Bath Temp. (°C)	Time (min)	Ultrasonic Time (min)	Cyanidin 3-Galactoside	Cyanidin 3-Glucoside	Cyanidin 3-Arabinoside	Cyanidin 3-Xyloside	Sum Avarage
1% citric acid								
50% ethanol	40	60	15	585.24 ± 70 ^a^	41.1 ± 7.9 ^a^	253.16 ± 23 ^a^	45.07 ± 5.24 ^ab^	924.57 ^c^
	50	60	30	572.27 ± 67 ^a^	36.91 ± 3.6 ^a^	262.69 ± 34 ^a^	46.40 ± 6.51 ^ab^	918.28 ^c^
	50	120	15	570.23 ± 64 ^a^	37.52 ± 4.9 ^a^	245.59 ± 15 ^a^	44.31 ± 4.35 ^ab^	897.66 ^abc^
	50	120	30	522.81 ± 10 ^a^	35.11 ± 0.99 ^a^	239.19 ± 4.9 ^a^	41.72 ± 0.95 ^ab^	766.64 ^abc^
	50	60	15	597.05 ± 41 ^a^	39.66 ± 3.0 ^a^	259.82 ± 5.1 ^a^	43.96 ± 2.31 ^ab^	938.66 ^abc^
	40	120	30	558.72 ± 21 ^a^	n.d.	220.53 ± 5.7 ^a^	48.30 ± 0.00 ^ab^	819.38 ^abc^
50% glycerol	60	60	30	578.88 ± 17 ^a^	35.14 ± 1.9 ^a^	229.37 ± 5.8 ^a^	54.05 ± 1.81 ^b^	799.77 ^abc^
1% formic acid								
50% ethanol	40	120	30	495.45 ± 17 ^a^	n.d	202.07 ± 2.9 ^a^	41.38 ± 0.00 ^a^	729.28 ^a^
	40	120	15	485.57 ± 61 ^a^	n.d	197.13 ± 23 ^a^	41.91 ± 3.89 ^ab^	716.41 ^a^
	50	60	30	527.48 ± 1 ^a^	n.d	211.95 ± 1.2 ^a^	45.94 ± 0.50 ^ab^	781.36 ^ab^

n.d. not detectable; CGaE: cyanidin-3-galactoside equivalent. The letters indicate significant differences between each cyanidin sample.

## Data Availability

Data will be provided by the corresponding authors exclusively on reasonable request, as they are for an ongoing study.

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
