# Peer review of "The Efficiency of Selected Green Solvents and Parameters for Polyphenol Extraction from Chokeberry (Aronia melanocarpa (Michx)) Pomace"

_foods, 2023, doi:10.3390/foods12193639_

Round 1

Reviewer 1 Report

This study investigated the impact of solvents, extraction temperature, and solvent pH level on polyphenol yield to explore optimal efficiency, which has the practical application value while lack of innovation.

There are some suggestions, which may be help.

1. The research investigated the effect of time on the extraction, but there were only two tested conditions, including different water bath of 60-120 min water bath time and ultrasonic treatment time of 15-30 min, the range of them needs to be expanded.

2. In Table 2. the unit of sonication temperature and time should be correctted.

3. In Figure1,2 the within-group significant difference should be analyzed among the extraction effects of the same extraction solvent, different extraction conditions and different extraction solvents, it will help to get clearer results and analysis.

4.The data in the research were mainly according to the results of HPLC, the chromatograms of HPLC in different operation conditions should be provided as Supplementary materials. The chromatograms of HPLC in Fig.3 maybe has some problem for it was so clean, which has almost no visible peaks of other components in the matrix. Could the purity get so high just after simple extraction processing?

This study investigated the impact of solvents, extraction temperature, and solvent pH level on polyphenol yield to explore optimal efficiency, which has the practical application value while lack of innovation.

There are some suggestions, which may be help for publication.

1. The research investigated the effect of time on the extraction, but there were only two tested conditions, including different water bath of 60-120 min water bath time and ultrasonic treatment time of 15-30 min, the range of them needs to be expanded.

2. In Table 2. the unit of sonication temperature and time should be correctted.

3. In Figure1,2 the within-group significant difference should be analyzed among the extraction effects of the same extraction solvent, different extraction conditions and different extraction solvents, it will help to get clearer results and analysis.

4.The data in the research were mainly according to the results of HPLC, the chromatograms of HPLC in different operation conditions should be provided as Supplementary materials. The chromatograms of HPLC in Fig.3 maybe has some problem for it was so clean, which has almost no visible peaks of other components in the matrix. Could the purity get so high just after simple extraction processing?

Author Response

For research article

Response to Reviewer 1 Comments

1. Summary

Thank you very much for taking the time to review this manuscript. Please find the detailed responses below and the corresponding revisions in track changes in the re-submitted files.

2. Questions for General Evaluation

Reviewer’s Evaluation

Response and Revisions

Does the introduction provide sufficient background and include all relevant references?

Yes/Can be improved/Must be improved/Not applicable

Thank you for your opinion and comments. We modified the article accordingly.

Are all the cited references relevant to the research?

Yes/Can be improved/Must be improved/Not applicable

Is the research design appropriate?

Yes/Can be improved/Must be improved/Not applicable

Are the methods adequately described?

Yes/Can be improved/Must be improved/Not applicable

Are the results clearly presented?

Yes/Can be improved/Must be improved/Not applicable

Are the conclusions supported by the results?

Yes/Can be improved/Must be improved/Not applicable

3. Point-by-point response to Comments and Suggestions for Authors

This study investigated the impact of solvents, extraction temperature, and solvent pH level on polyphenol yield to explore optimal efficiency, which has the practical application value while lack of innovation.

There are some suggestions, which may be help.

Comments 1: The research investigated the effect of time on the extraction, but there were only two tested conditions, including different water bath of 60-120 min water bath time and ultrasonic treatment time of 15-30 min, the range of them needs to be expanded.

Response 1: Thank you for your comment. In this paper, the effect of three tested conditions was evaluated. The examination of extraction method on different polyphenols and colour parameters of chokeberry belong to the wider PhD research work in our department. Based on the pre-experiment (did not published) we revealed that the waterbath temperature, time and sonication duration had mainly affect on the given parameters.

Comments 2: In Table 2. the unit of sonication temperature and time should be corrected.

Response 2: Thank you for remark. We have accordingly modified the unit in the Table 3 (page of 9).

Comments 3: In Figure1,2 the within-group significant difference should be analyzed among the extraction effects of the same extraction solvent, different extraction conditions and different extraction solvents, it will help to get clearer results and analysis.

Response 3: Thank you for your comments. The statistical analysis was expanded accordingly. (Page of 5-6, Table 2.)

Comments 4: The data in the research were mainly according to the results of HPLC, the chromatograms of HPLC in different operation conditions should be provided as Supplementary materials. The chromatograms of HPLC in Fig.3 maybe has some problem for it was so clean, which has almost no visible peaks of other components in the matrix. Could the purity get so high just after simple extraction processing?

Response 4: The HPLC chromatograms are provided in the Supplementary materials. Before the HPLC analysis, moreover during the extractions, the there was no purification steps. In our chokeberry samples there were only four main anthocyanin compounds at the wavelength of 520 nm, however one additional peak was detected around 3 minutes in different intensity.

4. Response to Comments on the Quality of English Language

Point 1: Minor editing of English language required.

Response 1. Thank you for your suggestion. English language was reviewed and made some changes in the text.

Reviewer 2 Report

The manuscript submitted for review concerns the use of green chemistry for the extraction of bioactive substances from plant raw materials. Today, this is a very popular trend and the research carried out fits into it. Nevertheless, the work does not show much originality apart from the raw material used, which is chokeberry fruit. The entire experiment was properly designed and described. However, I have great concerns about the correctness of the described profile obtained using the HPLC method.

1. Why did the authors limit themselves only to anthocyanins? Why didn't they analyze other derivatives such as phenolic acids and flavonols ?

2. I believe that the anthocyanin profile is misinterpreted. I have a lot of experience in working with these fruits and I know that the main profile of these compounds consists of 4 cyanidin derivatives: glucoside, galactoside, arabinoside and xyloside. In my opinion, the last peak, which is probably xyloside, should be quantified

Author Response

For research article

Response to Reviewer 2 Comments

1. Summary

Thank you very much for taking the time to review this manuscript. Please find the detailed responses below and the corresponding revisions in track changes in the re-submitted files.

2. Questions for General Evaluation

Reviewer’s Evaluation

Response and Revisions

Does the introduction provide sufficient background and include all relevant references?

Yes/Can be improved/Must be improved/Not applicable

Thank you for your opinion and comments. We have modified the article accordingly.

Are all the cited references relevant to the research?

Yes/Can be improved/Must be improved/Not applicable

Is the research design appropriate?

Yes/Can be improved/Must be improved/Not applicable

Are the methods adequately described?

Yes/Can be improved/Must be improved/Not applicable

Are the results clearly presented?

Yes/Can be improved/Must be improved/Not applicable

3. Point-by-point response to Comments and Suggestions for Authors

Comments 1: Why did the authors limit themselves only to anthocyanins? Why didn't they analyze other derivatives such as phenolic acids and flavonols ?

Response 1: Thank you for your question. This work is belong to the wider PhD research where we would like to analyze the mainly polyphenols in chokeberry after using different extraction methods. This paper is focusing on anthocyanins and colour parameters mainly while in our next experimental section the other polyphenols and antioxidants will examined further. To clarify the aim of this manuscript we have modified the text accordingly. (Page of 2, row 79-83)

Comments 2: I believe that the anthocyanin profile is misinterpreted. I have a lot of experience in working with these fruits and I know that the main profile of these compounds consists of 4 cyanidin derivatives: glucoside, galactoside, arabinoside and xyloside. In my opinion, the last peak, which is probably xyloside, should be quantified.

Response 2: Yes, we agree that the chokeberry consists of four main anthocyanins as we indicated earlier in the text also (page 8, row 255). The last, fourth peak is probably the cyanidin-3-xyloside based on the other studies, but we could not confirm the present of cyanidin-3-xyloside due to the absence of standards. However based on the HPLC analysis we quantified the last peak and results are inserted to the paper.(page of 10, Figure 3 and Table 4)

Reviewer 3 Report

Comments for the authors:

Abstract: The overall conclusion is missing stating the importance of the obtained results.

Introduction: What is the approx. quantity of the chokeberry waste remained after juice processing? 

Sample preparation: Moisture content of the dried material should be indicated.

Extraction procedure: Please, state that the UAE was performed in ultrasonic bath and include the ultrasound power. 

Table 1: I suggest adding codes to the extracts obtained at different extraction parameters.

Table 3: Statistical analysis should be included.

Author Response

For research article

Response to Reviewer 3 Comments

1. Summary

Thank you very much for taking the time to review this manuscript. Please find the detailed responses below and the corresponding revisions in track changes in the re-submitted files.

2. Questions for General Evaluation

Reviewer’s Evaluation

Response and Revisions

Does the introduction provide sufficient background and include all relevant references?

Yes/Can be improved/Must be improved/Not applicable

Thank you for your opinion and comments. We have modified the article accordingly.

Are all the cited references relevant to the research?

Yes/Can be improved/Must be improved/Not applicable

Is the research design appropriate?

Yes/Can be improved/Must be improved/Not applicable

Are the methods adequately described?

Yes/Can be improved/Must be improved/Not applicable

Are the results clearly presented?

Yes/Can be improved/Must be improved/Not applicable

3. Point-by-point response to Comments and Suggestions for Authors

Comments 1: Abstract: The overall conclusion is missing stating the importance of the obtained results.

Response 1: Thank you for your remark. The abstract section was completed by the overall conclusion. (page of 1, row 20-23).

Comments 2: Introduction: What is the approx. quantity of the chokeberry waste remained after juice processing? 

Response 2: It depends on the crushing and treating of the fruits and pressing method. Enzymatic mush treatment and use of a decanter during pressing: approx. 5-10% pomace. Application of enzyme treatment and horizontal basket hydraulic press: approx. 15-35% pomace. Enzymatic treatment, use of pneumatic presses: pomace can be up to 40-50%. Therefore, the utilization and further processing of this amount of material is very important in the food industry.

Comments 3.: Sample preparation: Moisture content of the dried material should be indicated.

Response 3: Thank you for this observation, the moisture content and water activity were measured and the results were shown in the text: “The moisture content of dried pomace was 11.77%, while the water activity decreased to 0.221.” (page of 3, row 102-103).

Comments 4: Extraction procedure: Please, state that the UAE was performed in ultrasonic bath and include the ultrasound power. 

Response 4: The power of ultrasound bath was inserted to the text: “For further extraction, samples were transferred to the ultrasonic (Bandelin, RK 52), and sonicated for 15 or 30 minutes at 35 kHz (as indicated in Table 1) to determine the optimal duration of the ultrasonic-assisted extraction.”(Page of 3, row 113)

Comments 5: Table 1: I suggest adding codes to the extracts obtained at different extraction parameters.

Response 5: Thank you for your suggestion. Unfortunately, we cannot add the suggested codes to the table since it only displays information on the chosen extraction temperature, duration, solvents, and their concentration, not the actual extracts. With the given parameters and solvents, a total of 36 extracts are generated, each with its own unique codes. However, adding these codes to the table would make it too large.

Comments 6: Table 3: Statistical analysis should be included.

Response 6: Thank you for your comments. The statistical analysis was expanded accordingly. (Page of 10, Table 4.)

Round 2

Reviewer 1 Report

The MS had revised according to the comments.

The MS had revised according to the comments.

Reviewer 2 Report

All my comments were positively received by the authors, which confirms the acceptance of the manuscript for publication